# The Role of MiR-181 Family Members in Endothelial Cell Dysfunction and Tumor Angiogenesis

**DOI:** 10.3390/cells11101670

**Published:** 2022-05-18

**Authors:** Chun Yang, Victor Passos Gibson, Pierre Hardy

**Affiliations:** 1Research Center of CHU Sainte-Justine, University of Montréal, Quebec, QC H3T 1C5, Canada; chun.yang.hsj@ssss.gouv.qc.ca; 2Departments of Pharmacology and Physiology, University of Montréal, Quebec, QC H3T 1C5, Canada; victor.passos.hsj@ssss.gouv.qc.ca; 3Departments of Pediatrics, University of Montréal, Quebec, QC H3T 1C5, Canada

**Keywords:** angiogenesis, miR-181, tumor angiogenesis, endothelial dysfunction, BBB/BTB, anti-angiogenic therapy, nanoparticles

## Abstract

Endothelial dysfunction plays a critical role in many human angiogenesis-related diseases, including cancer and retinopathies. Small non-coding microRNAs (miRNAs) repress gene expression at the post-transcriptional level. They are critical for endothelial cell gene expression and function and are involved in many pathophysiological processes. The miR-181 family is one of the essential angiogenic regulators. This review summarizes the current state of knowledge of the role of miR-181 family members in endothelial cell dysfunction, with emphasis on their pathophysiological roles in aberrant angiogenesis. The actions of miR-181 members are summarized concerning their targets and associated major angiogenic signaling pathways in a cancer-specific context. Elucidating the underlying functional mechanisms of miR-181 family members that are dysregulated in endothelial cells or cancer cells is invaluable for developing miRNA-based therapeutics for angiogenesis-related diseases such as retinopathies, angiogenic tumors, and cancer. Finally, potential clinical applications of miR-181 family members in anti-angiogenic tumor therapy are discussed.

## 1. Introduction

Angiogenesis refers to the formation of new blood vessels from existing vessels and is involved in endothelial cell (EC) proliferation, migration, and morphogenesis. Angiogenesis naturally occurs in an organism during development and physiological processes such as wound healing and the menstrual cycle [1]. It is a dynamic process tightly regulated spatially and temporally among angiogenic factors, extracellular matrix components, and ECs [2]. Angiogenesis is controlled by a precise balance between stimulatory and inhibitory angiogenic factors in healthy tissues [3]. Among the many pro-angiogenic factors, vascular endothelial growth factor (VEGF) and angiopoietins (Angs) have emerged as the primary regulators with specificity for ECs. Binding to their receptors on ECs, these factors are responsible for the proliferation, EC migration, and the integrity and maintenance of the vascular network [4]. Fibroblast growth factor (FGF), platelet-derived growth factor (PDGF), transforming growth factor-β (TGF-β), hepatocyte growth factor, and matrix metalloproteinases (MMPs) are also known to drive vascular growth [5] Anti-angiogenic factors include angiostatin, endostatin, thrombospondin-1, interferon-γ, anti-factor VIIa, and tissue inhibitors of MMPs (TIMPs). Abnormal angiogenesis occurs when this balance is disturbed and is associated with human pathologies such as cancer, retinopathies, and ischemic diseases [5].

Recent studies revealed that functional non-coding RNAs (ncRNAs) such as microRNAs (miRNAs), circular RNAs (circRNAs), small nucleolar RNAs, and long ncRNAs (lncRNAs) play critical roles in angiogenesis [6]. For example, lncRNAs are a group of more than 200 nt ncRNAs involved in many cellular processes, some of which are dysregulated in pathological angiogenesis. LncRNAs regulate translation by direct binding to target mRNAs, competing endogenous RNAs, and sponging miRNAs [7]. CircRNAs are a class of endogenous ncRNA characterized by their unique structure with a 3′ to 5′ end-joining event [8]. CircRNA expression is cell type- and tissue-specific; they sponge miRNAs to regulate their functions [9]. This review focuses on miRNAs in the context of pathological angiogenesis.

MiRNAs are endogenous, highly conserved, double-stranded, ncRNA molecules of about 20–22 nucleotides in length in their mature form. MiRNAs originally reside in the nucleus and are transcribed by RNA polymerase II into a long primary molecule (pri-miRNA), which is further cleaved into approximately 70–100 nucleotide hairpin-shaped precursors miRNA (pre-miRNA) by RNA-specific RNase III type endonuclease, Drosha, and its cofactor, DiGeorge syndrome critical region (8DGCR8) [10]. The pre-miRNA is actively transported to the cytoplasm via an exportin-5- and ran-GTP-dependent mechanism. Once in the cytoplasm, pre-miRNAs are cleaved into ~22-nt duplex mature miRNA duplexes by dicer, a cytoplasmic double-stranded ribonuclease. Then, the miRNA duplex is integrated into the effector RNA-induced silencing complex (RISC), from which a mature single-stranded miRNA is produced and binds to the 3′-untranslated region (3′-UTR) of target mRNAs to block translation or cleave mRNA. The biogenesis of microRNA was reviewed by Bartel et al. [11]. Many studies suggested the involvement of miRNAs in angiogenesis. Dicer and drosha are necessary for miRNAs expression in ECs and are essential in angiogenesis. Knocking down enzymes reduces capillary development and tubule-forming activity in ECs [12,13]. Many miRNAs exhibit striking organ- or tissue-specific expression patterns; some are found abundantly in angiogenesis-associated ECs and regulate blood vessel development and angiogenesis [14,15,16]. Recently, the roles of miRNAs in tumor angiogenesis have also been extensively investigated [7].

## 2. MiR-181 Family Members

The miR-181 family consists of four highly conserved members: miR-181a, miR-181b, miR-181c, and miR-181d [17]. They are independently derived from six precursors located on three chromosomes. MiR-181a/b-1 is located on chromosome 1, miR-181a/b-2 is located on chromosome 9, and miR-181c/d clusters on chromosome 19 [18]. Although identical in their mature sequences, miR-181b-1 and miR-181b-2 are located on different chromosomes [19]. No protein-coding regions are found within the chromosomal regions of the miR-181 transcripts, suggesting these miRNAs are transcribed independently. The -3p and -5p strands of each miR-181 family member differ by only a few nucleotides. However, each miR-181 member targets different genes, leading to various functionality and context-sensitive activities [20]. The sequence homology and difference among miR-181a-d, and their gene loci on different chromosomes were elucidated by Indrieri et al. [21]. MiR-181s are aberrantly expressed in tumor tissues and exhibit oncogenic or tumor-suppressive properties in a cancer-specific manner [22,23].

## 3. MiR-181 in EC Differentiation

The specification of arterial, venous, and lymphatic EC fate is critical during vascular development. Several studies demonstrated the involvement of miR-181 family members in embryonic vascular development and endothelial differentiation. Kazenwadel et al. found higher expression of miR-181a in embryonic blood vascular ECs than lymphatic ECs [24]. The expression of prospero homeobox protein 1 (prox1), a crucial gene for the specification and maintenance of lymphatic EC identity, is directly suppressed by miR-181a. Importantly, increased miR-181a activity in primary embryonic lymphatic ECs resulted in the reprogramming of lymphatic ECs toward a blood vascular phenotype by targeting Prox1, suggesting that miR-181a is critical for the EC fate during vascular development and neo-lymphangiogenesis [24]. Shaik et al. studied the role of miRNAs in the differentiation of adipose-derived stromal/stem cells (ASCs) into the endothelial lineage [25]. They found that miR-181a-5p is upregulated during endothelial differentiation. Although the specific mechanisms of miR-181a-5p function in ASCs are unclear, the overexpression of miR-181a-5p likely induces endothelial differentiation in ASCs [25]. Pluripotent human embryonic stem cells (hESCs) can be directly differentiated toward EC lineages and express a unique panel of miRNAs critical in EC differentiation and developmental angiogenesis [16,26]. Kane et al. investigated the functional role of miRNAs in hESC differentiation to vascular ECs [27]. They observed that miR-181a and miR-181b (miR-181a/b) are increased to a peak in mature hESC-ECs and adult ECs. High levels of miR-181a/b enhanced the expression of EC-specific markers (pecam1 and VE-cadherin) accompanying differentiation of hESCs to vascular ECs. In vivo, overexpression of miR-181b improved hESC-EC-induced therapeutic neovascularization angiogenesis in a mouse model of limb ischemia, indicating that miR-181b is capable of potentiating EC differentiation from pluripotent hESCs and a direct cell effect of transplanted cells may contribution to the in vivo neovascularization.

These data strongly suggest the essential role of miR-181a and miR-181b in embryonic vascular development and inducing embryonic stem cell differentiation toward EC (Table 1). This mechanism suggests a potential strategy for optimizing the generation of pro-angiogenic vascular ECs for regenerative medicine.

## 4. MiR-181 in EC Barriers of the Blood-Brain Barrier (BBB) and Blood-Tumor Barrier (BTB)

EC layers of blood and lymphatic capillaries separate the blood or lymph fluids from the parenchymal tissues in humans. The barrier elements in the central nervous system (CNS) are the BBB, the cerebrospinal fluid barrier, and the meningeal barrier [32]. The BBB is composed of a non-fenestrated layer of ECs of the cerebral microvasculature, which is held together by tight junctions [33]. Tight junctions (also known as zona occludens) are multiprotein junctional complexes containing peripheral membrane proteins zonula occludins (ZO) such as ZO-1, and transmembrane proteins such as occludin, claudins, and junction adhesion molecule [34,35]. Although properties of the BBB are manifested within ECs, the interactions of ECs with pericytes, microglia, astrocytes, neural cells, and immune cells are essential for the control of vascular permeability and CNS functions [33,36]. Recent genetic and molecular studies demonstrated the essential role of canonical Wnt signaling in forming tight junction proteins and driving human ECs to develop distinct BBB features in the CNS [37,38]. Like the BBB, the BTB is composed of microvascular ECs interconnected by tight junctions that restrict paracellular diffusion and limit the delivery of chemotherapeutic drugs to tumor tissues [39,40]. The BTB is more permeable than the BBB; however, the structural integrity of BTB varies between metastatic lesions and tumor types. The differences between BBB and BTB were reviewed by Arvanitis et al. [41]. Brain metastasis–the migration of cancer cells through the destruction of the BBB/BTB–is associated with poor outcomes [42].

In human glioma, miR-181a and miR-181d are critical miRNAs in opening BTB by impairing the expression of tight junction proteins. Ma et al. found that miR-181a expression is upregulated in glioma endothelial cells (GECs) [28]. Overexpression of miR-181a increased permeability of BTB by direct targeting Kruppel-like factor 6 (KLF6) and downregulating tight junction proteins. KLF6 is a transcription factor of the zinc-finger family that regulates EC motility by upregulating the promoter activities of tight junction proteins ZO-1, occludin, and claudin-5 in GECs [28,43]. Guo et al. reported that miR-181d-5p directly suppressed the expression of sex-determining region Y-box protein 5 (SOX5) in GECs to increase BTB permeability [31]. The transcriptional factor SOX5, which determines cell fate and differentiation, is highly expressed in glioma and GECs [44,45]. SOX5 binds to the promoters of the tight junction proteins ZO-1, occludin, and claudin-5 to activate their expression. Notably, the miR-181d-5p activity can be abrogated by lncRNA nuclear paraspeckle assembly transcript 1 (NEAT1), which is upregulated in GECs [31]. Two independent studies demonstrated the effect of miR-181c in promoting BBB permeability. Tominaga et al. found that miR-181c is significantly enriched in cancer-derived extracellular vesicles (EVs), which can be incorporated into ECs to trigger the breakdown of BBB and regulate the invasion through the BBB of cancer cells [29]. Overexpression of synthetic miR-181c in ECs promoted the destruction of the BBB in an in vitro model that consisted of primary cultures of brain capillary ECs, brain pericytes, and astrocytes. Mechanistically, miR-181c directly downregulates 3-phosphoinositide-dependent protein kinase-1 (PDPK1) in ECs, leading to the reduction of phosphorylated cofilin and the localization of tight junction proteins–N-cadherin and actin [29]. Yu et al. discovered the involvement of miR-181c-5p in inducing blood-spinal cord barrier (BSCB) permeability under hypoxic conditions [30]. In a hypoxic BSCB model formed by coculture of spinal cord astrocytes and primary rat brain microvascular ECs, overexpression of miR-181c-5p caused BSCB integrity disruption and downregulation of tight junction protein expression. This study also suggested the downregulation of SOX5 expression of SOX5 by miR-181c-5p, which contributed to the dysregulation of tight junction proteins expression.

In summary, miR-181 family members miR-181a, miR-181c, and miR-181d induce BBB/BTB permeability in pathological conditions such as brain malignancies or hypoxia (Table 1), revealing potential therapeutic targets for the treatment of brain gliomas [28].

## 5. The Pathophysiological Role of MiR-181 in EC Angiogenesis

The anti-angiogenic activities of miR-181a and miR-181b were suggested in several studies. Zhu et al. showed that lncRNA H19 possesses pro-angiogenic activity in human microvascular endothelial cells (HMEC-1) by downregulation of miR-181a [46]. Overexpression of miR-181a eliminated the pro-angiogenic effects of H19 by abolishing H19-induced expression of critical regulators of angiogenesis such as MMP-2, MMP-9, VEGF, and eNOS; H19-activated JNK and AMPK signaling [46,47]. Sun et al. reported that miR-181a-5p and miR-181b-5p (miR-181a/b-5p) suppressed HUVECs angiogenesis by direct targeting platelet-derived growth factor-alpha (PDGFA), a protein responsible for angiogenesis, migration, and invasion [48]. Li et al. found that miR-181b expression was downregulated in hypoxia-stimulated primary HUVECs, which fostered the initiation and maintenance of angiogenesis [49]. Overexpression of miR-181b suppressed angiogenesis in vitro by directly suppressing the targeting of cellular communication network factor 1 (CCN1) and inhibiting the AMPK signaling pathway. Moreover, injection of agomir-181b impaired perfusion recovery in mouse hindlimb ischemia model and capillary density in a Matrigel plug assay, suggesting that miR-181b inhibited capillary outgrowth. Cui et al. provided evidence that sodium arsenite (an environmental toxin) induced developmental toxicity and reduced the expression of miR-181b in human vascular ECs [50]. Overexpression of miR-181b inhibited arsenic-induced EC migration and in vitro angiogenesis by directly targeting neuropilin-1 (NRP1) [50], a transmembrane glycoprotein that plays essential roles in angiogenesis and vascular development [51,52].

Two independent studies strongly supported the pathophysiological role of miR-181a in the inhibition of retinal neovascularization (RNV), the most common pathological angiogenesis-related retinopathy, including retinopathy of prematurity (ROP, a proliferative retinal vascular disease, a significant cause of childhood blindness), exudative age-related macular degeneration, and proliferative diabetic retinopathy [53,54]. MiR-181a is highly expressed in the retina and choroidal tissues [55]. We demonstrated the role of miR-181a in the regulation of ocular neovascularization in various pathophysiological conditions [56]. In addition to the substantial anti-angiogenic effect of miR-181a in ex vivo choroidal neovascularization and RNV in an in vivo mouse model of ROP, angiogenesis-related genes such as mitogen-activated protein kinase 1 (MAPK1), B-cell lymphoma 2 (Bcl2), and VEGF were downregulated in miR-181a-5p-overexpressed ECs [57,58,59]. In agreement with our observation, Chen et al. found that miR-181a-5p was enriched in the normal mouse retina but reduced in the retina of a mouse model of ROP [60]. Intraocular injection of miR-181a-5p resulted in the suppression of RNV in vivo. The expression of endocan, a direct target of miR-181a-5p, was strongly upregulated in the retina in oxygen-induced retinopathy. Endocan is specifically expressed in ECs and highly enriched in retinal endothelial tip cells, where it participates in EC activation and angiogenesis [61,62,63]. The authors reported that miR-181a-5p inhibited RNV by directly repressing endocan and suppressing VEGF-mediated extracellular signal-regulated protein kinase 1 and 2 (ERK1/2) pathway activation.

In contrast to the aforementioned anti-angiogenic effect of miR-181a/b, Zhao et al. suggested a pro-angiogenic role of miR-181a in HUVECs [64]. They found that miR-181a expression was regulated by RNA binding protein fragile X mental retardation protein (FMRP), which participates in pri-miRNA processing and functions as a translational repressor [65]. The upregulated miR-181a in HUVECs was accompanied by increased HUVECs angiogenesis and decreased ubiquitous Ca^2+^ binding protein calmodulin (CaM). As the calcium/calmodulin-dependent protein kinase II (CaMKII) plays a crucial role in triggering VEGF expression and endothelial angiogenesis [66,67], miR-181a might promote angiogenesis by targeting the CaM-CaMKII pathway in HUVECs [64]. Moreover, miR-181b-5p in exosomes derived from adipose-derived stem cells had pro-angiogenic activity in primary rat brain microvascular endothelial cells (BMECs) [68]. Yang et al. showed that the exosomal miR-181b-5p promoted the mobility and angiogenesis of BMECs after oxygen-glucose deprivation, and this effect was a consequence of directly targeting the transient receptor potential melastatin 7 (TRPM7), leading to increased HIF-1α and VEGF, and decreased tissue inhibitor of metalloproteinase 3 (TIMP3) [69]. TRPM7 has an essential role in the adhesion and tube-formation of vascular ECs [70].

A pro-angiogenic role of miR-181c was demonstrated in several pathophysiological conditions. Yu et al. showed that overexpression of miR-181c-5p reversed the lncRNA SNHG1-suppressed angiogenesis of bone marrow-derived endothelial progenitor cells (BM-EPCs) by downregulating secreted frizzled-related protein-1 (SFRP1) [71]. This sequence activates the Wnt/β-catenin pathway, a crucial cellular signal transduction pathway controlling angiogenesis and vasculogenesis [72,73]. Deng et al. found that miR-181c acted as an angiogenesis inducer in retinal angiogenesis of ROP [74]. The circular RNA circPDE4B is abundantly expressed in the human retina and plays a role in retinal development and function [75]. CircPDE4B was downregulated in ROP models with reduced capability of sponging miR-181c; hence, it suppressed the expression of the tumor suppressor protein von Hippel-Lindau (VHL) in hypoxia-induced human retinal ECs and subsequently activated the expression of HIF-1α and VEGFA and HIF-1α-dependent angiogenesis [74,76]. Song et al. reported upregulated miR-181c in the ECs of endometrium in patients with endometriosis [77]. Endometriosis is a multifactorial disease in which angiogenesis plays an essential role [78]. In the ECs of endometriosis, miR-181c targeted TIMP3, a matrix metalloproteinase (MMP) inhibitor and angiogenesis suppressor, by binding to endothelial growth factor receptor 2 [79]. MiR-181c suppression of TIMP3 expression may contribute to endometriosis by regulating cell dysfunction and angiogenesis.

Regarding the angiogenic role of miR-181d, Zhang et al. observed that miR-181d-5p attenuated the pro-angiogenic capacity of lncRNA nuclear paraspeckle assembly transcript 1 (NEAT1) in oxidative stress-induced HUVECs [80]. MiR-181d-5p suppressed cyclin-dependent kinase inhibitor 3 (CDKN3) expression in HUVECs to inactivate the Akt signaling pathway.

In conclusion, although contradictory functions of miR-18 family members found in the referred experiment settings ((Figure 1, Table 2), the antiangiogenic role of miR-181a and miR-181b, and pro-angiogenic role of miR-181c have been revealed in in vivo animal models [49,56,60,74]. Unlike the in vitro cell culture studies which tend to be more variable and prone to artifacts, the in vivo studies are more relevant to the disease state.

## 6. MiR-181 in Tumor Angiogenesis

Rapidly proliferating cancer cells urgently demand continuous blood and nutrient supply in the tumor microenvironment. As tumor mass increases, inner tumor cells become relatively hypoxic, upregulating the expression of angiogenic growth factors to induce the proliferation and migration of ECs for angiogenesis [81]. Tumor angiogenesis aids tumor progression and metastasis by providing oxygen and nutrients from newly formed blood vessels and physical routes for migration toward metastasis sites (2). Many pro-angiogenic factors are upregulated in tumor cells and tumor-associated stromal cells, including VEGF, FGF, and delta ligand-like 4 (Dll4). Because of the high expression of VEGF and other angiogenic factors in the tumor environment, tumor-associated capillaries are malformed, highly permeable, and leaky [82]. Recent studies indicated the involvement of miR-181 family members in the regulation of tumor angiogenesis (Figure 2, Table 3).

Matrix metalloproteinase 14 (MMP-14) upregulation is often observed in invasive human cancers and correlates with the aggressiveness of human cancer cell lines [83,84]. MiR-181a-5p was downregulated in aggressive human breast cancer, fibrosarcoma, and colon cancer, where the MMP-14 is upregulated. MMP-14 is a direct target gene of miR-181a-5p; essentially, miR-181a-5p-mediated reduction of MMP-14 was sufficient to reduce in vivo angiogenesis in chick chorioallantoic membrane assays and impair new blood vessel formation induced by HT1080 (a human fibrosarcoma cell line) [85]. The tumor angiogenesis inhibitory effect of miR-181a-5p was observed in hepatocellular carcinoma and invasive bladder cancer. Mechanistically, miR-181a-5p targets endocan, the levels of which in ECs from hepatocellular carcinoma or invasive bladder cancer were associated with angiogenesis and tumor invasion [86,87].

MiR-181a exhibiting pro-tumor angiogenesis effects were found in several cancers. Adenosquamous carcinoma of the pancreas (ASCP) is an uncommon exocrine pancreatic malignancy with poor outcomes [15]. Although ASCP is characterized by a low microvascular density and collapsed vasculature, high levels of miR-181a-5p, VEGFA, HIF1α, and angiopoietin-1 (Ang-1) were detected in ASCP. A hypoxic environment (hypoxia or HIF-1α transfection) induced miR-181a-5p to function as an angiogenesis promoter through upregulation of VEGF in cancer cells [15,88]. In papillary thyroid cancer (PTC), a miR-181a mimic and hypoxia-induced exosomal miR-181a augmented the proliferation, migration, and capillary-like network formation of HUVECs [89]. Hypoxic PTC-secreted exosomal miR-181a promoted angiogenesis and tumor growth in PTC tumor xenografts. Mechanistically, miR-181a directly inhibited histone-lysine N-methyltransferase-3 (MLL3) and downregulated the disheveled binding antagonist of beta-catenin 2 (DACT2), which upregulated YAP and VEGF in HUVECs. MLL3 belongs to the mixed-lineage leukemia (MLL) family and functions as an H3K4me1 methylase. Downregulation of MLL3 reduced the expression of tumor suppressor DACT2 and promoted PTC metastasis by activating Wnt signaling [90,91].

In colorectal cancer (CRC)–a very aggressive metastatic tumor with active angiogenesis–miR-181a induced tube-formation in ECs and promoted angiogenesis in vivo by directly inhibiting SRC kinase signaling inhibitor 1 (SRCIN1, also known as p140Cap) [92]. SRC kinase promotes angiogenesis via the STAT3-VEGF pathway [93]. Downregulation of SRCIN1 by miR-181a led to VEGF secretion and activation of angiogenesis in CRC [92,94]. MiR-181a promoted chondrosarcoma angiogenesis and metastasis [95].

Chondrosarcoma is the most common primary malignant bone tumor in adults. Patients with this disease have poor survival. Sun et al. found that miR-181a and VEGF are highly expressed in human chondrosarcomas and a derived cell line [88]. Hypoxia and overexpression of HIF-1α increased miR-181a expression and induced expression of VEGF in chondrosarcoma cells [88]. A study found that the regulator of G-protein signaling 16 (RGS16) is a direct target of miR-181a [95]. RGS16 is a negative regulator of CXC chemokine receptor 4 (CXCR4) signaling associated with enhanced VEGF and MMP1 expression [96,97,98]. Therapeutic inhibition of miR-181a decreased expression of VEGF and MMP1 and suppressed angiogenesis in xenograft tumors, suggesting that targeting miR-181a can be a possible antagomir-based therapy in chondrosarcoma [95].

**Table 3 cells-11-01670-t003:** Roles of miR-181 in tumor angiogenesis.

Activity in Tumor Angiogenesis	MiR-181	Upstream Events	Targets	Downstream Signaling/Events	Ref
inhibits angiogenesis in breast cancer, colon cancer, fibrosarcoma	miR-181a-5p		MMP-14		[85]
inhibits angiogenesis in bladder cancer and hepatocarcinoma	miR-181a-5p		endocan	VEGF signaling↓	[86,87]
promotes angiogenesis in ASCP	miR-181a-5p			VEGF signaling↑	[15]
promotes angiogenesis in PTC	exosomal miR-181a	hypoxic	MLL3	DACT2↓, YAP-VEGF signaling↑	[89]
promotes angiogenesis in CRC	miR-181a		SRCIN1	SRC-VEGF pathway↑	[92]
promotes angiogenesis in chondrosarcoma	miR-181a		RGS16	CXCR4 signaling↑, VEGF↑, MMP1↑	[88,95]
promotes angiogenesis in ESCC	miR-181b-5p		PTEN, PHLPP2	Akt signaling↑	[99]
promotes angiogenesis in Rb	miR-181b	hypoxia	PDCD10, GATA6		[100]
inhibits angiogenesis in renal carcinomas	miR-181b		FGF1, PLAU	FGFR signaling↓	[101]
inhibits angiogenesis in lung cancer	miR-181b-3p			EC sprouting↓	[102]

Similar to miR-181a, miR-181b exhibited pro- and anti-angiogenic effects in different cancers. Wang et al. showed that tumor-derived extracellular vesicles-encapsulated miR-181b-5p (EVs-miR-181a-5p) induced angiogenesis in esophageal squamous cell carcinoma (ESCC) [99]. High levels of miR-181b-5p in ESCC tissues and serum EVs of ESCC patients correlated with poor outcomes. ESCC-derived EVs-miR-181b-5p can be transferred to vascular ECs and promotes angiogenesis to foster metastasis of ESCC. MiR-181b-5p inhibits phosphatase and tensin homolog (PTEN) and PH domain and leucine-rich repeat protein phosphatases 2 (PHLPP2), leading to the activation of Akt signaling to promote angiogenesis [99]. Hypoxia-induced miR-181b promoted the angiogenesis of retinoblastoma (Rb, a malignancy originating from the retina) by downregulating programmed cell death-10 (PDCD10) and GATA binding protein 6 (GATA6) [100]. MiR-181a is endogenously expressed in retina cells and is upregulated under hypoxic conditions in Rb cells [103]. PDCD10 and GATA6 are direct targets of miR-181b. PDCD10 is expressed in ECs and functions as an anti-angiogenic transcription factor in vascular development/maturation [104,105]. GATA6 is essential in regulating angiogenic function and EC survival [106].

In contrast to the pro-angiogenic role of ESCC-derived EVs, human liver stem cell (HLSC)-derived EVs inhibited tumor angiogenesis. Lopatina et al. observed that HLSC-derived EVs inhibited the angiogenic properties of tumor-derived ECs isolated from surgical specimens of patients with renal carcinomas [101]. MiR-181b in HLSC-EVs exhibited strong angiogenic inhibitory properties. When upregulated in tumor ECs treated with HLSC-EVs, miR-181b suppressed the expression of the pro-angiogenic genes fibroblast growth factor 1 (FGF1) and urokinase-type plasminogen activator (uPA). FGF1 is an essential pro-angiogenic factor involved in tumor angiogenesis in a VEGF-independent manner [107,108]; uPA is critical for tumor angiogenesis and progression by participating in the proteolytic processes of extracellular matrix degradation [109]. Along these lines, Green et al. observed that miR-181b-3p is upregulated in ECs in lung cancer [102]. An in vitro study showed that overexpression of miR-181b-3p reduced endothelial sprouting, suggesting an anti-angiogenesis role of miR-181b-3p in lung cancer.

The studies mentioned above provide evidence that the dysregulated expression of miR-181 family members may affect the activity of ECs and lead to abnormal angiogenesis, disruption of the BBB/BTB, and increased permeability (Table 1).

MiR-181 family members may play important roles in tumor angiogenesis, invasion, and metastasis by regulating multiple target genes related to these phenotypes. These studies also suggest that the role of individual miR-181 family members in promoting or inhibiting tumor angiogenesis is tissue- and cancer cell type-specific. Knowledge about the pathophysiological roles of miRNA-181 in abnormal angiogenesis provides possible future therapeutic interventions for angiogenesis-related diseases.

## 7. Potential Clinical Application of AngiomiR-Based Cancer Therapy

Tumor angiogenesis is a fundamental step in tumor growth and metastasis. Angiogenesis is essential for other biological processes, and abnormal pathological angiogenesis is a common denominator of angiogenesis-related diseases; therefore, angiogenesis regulators have immense therapeutic potential that exceeds cancer therapy. In the last few decades, the Food and Drug Administration approved several angiogenesis inhibitors and anti-angiogenetic agents such as monoclonal antibodies that block VEGF and VEGF receptors, or tyrosine kinase-specific inhibitors for the treatment of several human malignancies [110]. However, results were disappointing concerning therapeutic efficacy. The anti-angiogenic monotherapies delivered minimal clinical benefits, which may have resulted from resistance caused by the activation of redundancy of angiogenic factors produced by tumor cells that sustain tumor vascularization and growth [111]. These findings highlight the need to identify alternative agents and complementary approaches. In recent years, miRNAs have emerged as modulators of angiogenesis. These molecules regulate several target genes that may potentially overcome compensation mechanisms. For these reasons, angiomiRs have attracted substantial attention as targets for developing novel anticancer drugs.

In a therapeutic context, to restore miRNAs downregulated in ECs or cancer cells, administration of miRNA-specific mimics (synthetic oligonucleotides) can re-establish miRNA levels to their basal non-pathological states and restore their biological functions. Conversely, to suppress the high aberrant expression, several approaches can be used, including anti-miRNA oligonucleotides (AMOs or antimiRs, to silence endogenous miRNAs), miRNA sponges, and genetic knockouts using CRISPR/Cas9 genome-editing technology [112,113,114,115]. Despite their therapeutic potential, miRNAs have several drawbacks for therapeutic development, including potential cytotoxic effects on healthy tissues, low stability, low endocytosis, and immunotoxicity of the therapeutic miRNA [116]. Chemical modifications and nanotechnology-based delivery systems have been developed to encapsulate therapeutic miRNAs [117]. Virus-like nanoparticles and non-viral nanoparticles (NPs) have successfully delivered miRNA mimics or antagonists [118]. These NPs include liposomes, polymers, dendrimers, nanocelles, solid lipid NPs, silica NPs, nanotubes, metal NPs, and quantum dots [115]. Lipid nanoparticles (LNPs) are the gold standard delivery agent for nucleic acids [119]. LNPs closely pack oligonucleotides through electrostatic interaction, protecting them from nuclease degradation and overcoming physiological barriers that hinder nucleic acid translation [120]. LNPs efficiently target the liver or other organs by facile modulation of lipid components [121]. A milestone in gene therapy occurred in 2018 when LNPs allowed the translation of the first FDA-approved RNA interference (RNAi)-based drug (Onpattro, Patisiran) [122]. A few years saw the breakthrough achievement of mRNA vaccines against coronavirus disease 2019 (COVID-19). An LNP-platform encapsulating miRNA to treat solid tumors reached a phase 1 trial (NCT04675996) [123]. It is expected that non-viral vectors such as LNPs successfully promote the translation of angiomiR by delivering the cargo to specific angiogenesis-promoting cells and highly angiogenic tumors.

NPs can be modified to increase nucleic acid stability in serum and improve their endosomal escape and biocompatibility. Surface functionalization of NPs with a targeting moiety to surface-expressed biomolecules found either on ECs or the tumor cells can increase selectivity and specific uptake in the target tissue (active targeting) [124,125]. Likewise, modulation of NP components with stimuli-responsive materials can promote selective delivery of cargo in the tumor microenvironment [125]. Fine-tuning of physical-chemical properties can also influence the in vivo fate of NPs. NPs are 50–200 nm in diameter and are therefore expected to achieve retention in tumor tissues; 20–50 nm NPs are suitable for drug delivery through the BBB [126]. Compared to negatively charged or neutral NPs, positively charged NPs can enter cells but are more toxic [127].

Several pre-clinic studies showed promise for potential clinical applications of miRNA-based nanocarriers targeting tumor angiogenesis. For example, van Zandwijk et al. reported the successful use of targeted minicells loaded with miR-16 mimic for treating patients with recurrent malignant pleural mesothelioma and non-small cell lung cancer [128]. AngiomiR-16 is a tumor suppressor that affects cancer cell growth, migration, and angiogenesis. The authors also suggested the urgency of combining miRNA-based nanotechnology with chemotherapy or immune checkpoint inhibitors. It is noteworthy that EVs or exosomes gained potential use as drug delivery vehicles. This phenomenon occurs because EVs and exosomes exhibit less immunogenicity and are less toxic than other artificial delivery NPs [129]. They can cross the plasma membranes and diffuse in tumor tissues. Mesenchymal stromal cell (MSC)-derived exosomes as carriers for angiomiR-146b delivery altered glioblastoma development in the animal brain [130].

NP delivery platforms successfully delivered AMOs against miR-181a in a preclinical chondrosarcoma model [131]. MiR-181a is highly expressed in chondrosarcoma and functions as an oncomiR to promote tumor angiogenesis and metastasis [95]. Sun et al. conducted a systemic treatment with AMO directed against miR-181a using a nanopiece delivery platform (NPIs/anti-miR-181a) [131]. NPIs are nontoxic and penetrate the negatively charged cartilaginous matrix. NPIs/anti-miR-181a showed high transfection efficiency in chondrosarcoma cells, and systemic delivery of NPs/anti-miR-181a restored the tumor suppressor gene RGS16 expression, decreased tumor volume, and prolonged survival. A tumor-suppressor role of miR-181a-5p in Rb was revealed in our previous studies [132]. We also demonstrated that lipid NPs encapsulated with miR-181a-5p mimic exhibited strong inhibitory effects on Rb cell viability, and NPs loaded with miR-181a-5p and melphalan (a chemotherapeutic drug) exhibited a complementary anti-Rb effect.

## 8. Conclusions

Angiogenesis is a significant determinant of cancer progression and other vascular-dependent diseases. MiRNA-based approaches represent compelling solutions because they regulate multiple target genes. Because chemotherapy is angiogenesis-dependent, miRNAs targeting ECs may overcome chemotherapy resistance. In the last decade, fundamental research into the roles of miR-181 family members in pathological angiogenesis (including tumor angiogenesis) has been increasing significantly. These molecules are master regulators of pathological angiogenesis by interfering with critical angiogenic factors and signaling pathways. MiR-181 members might play conflicting roles in EC angiogenesis depending on the type of tumor or cell type. Despite a lack of evidence for the potential therapeutic applications of miR-181 members in patients, there is no doubt that an in-depth understanding of the biology and pathological function of miR-181 members in tumors and ECs is critical for translating this knowledge to the bedside. Appropriate delivery vehicles facilitate safe and efficient systemic delivery, and the therapeutic potential of miR-181 therapeutics for patients with angiogenesis-related diseases will eventually materialize.

## Figures and Tables

**Figure 1 cells-11-01670-f001:**
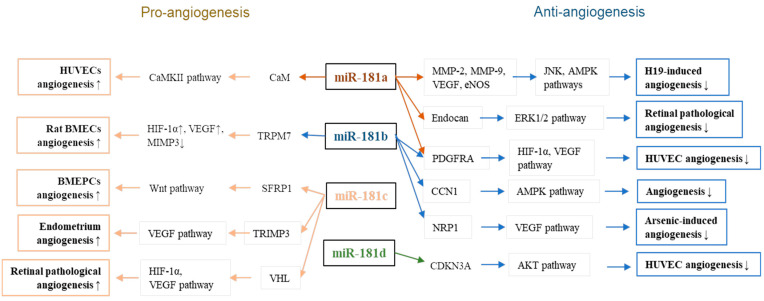
Roles of miR-181 family members in endothelial cell angiogenesis.

**Figure 2 cells-11-01670-f002:**
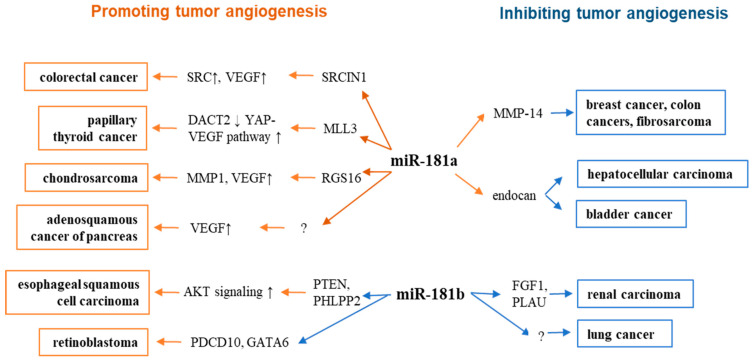
Role of miR-181 family members in angiogenesis of different types of cancer.

**Table 1 cells-11-01670-t001:** Roles of miR-181 family members in EC differentiation and EC barrier permeability.

Function	MiR-181 Members	Upstream Events	Target	Downstream Signaling/Events	Ref
EC differentiation					
promotes lymphangiogenesis toward vascular EC development	miR-181a		Prox1	ERK1/2 pathway	[24]
induces endotheliogenesis in ASCs	miR-181a-5p			VEGF, VWF, CD31	[25]
enhances hESCs differentiation to vascular ECs	miR-181a/b			Pecam1, nitric oxide VE-Cadherin	[27]
BBB/BTB					
increases BTB permeability	miR-181a		KLF6	tight junction proteins	[28]
increases BBB permeability	miR-181c		PDPK1	cofilin phosphorylation	[29]
increases blood-spinal cord barrier (BSCB) permeability	miR-181c-5p	NHO-1	SOX5	tight junction proteins	[30]
increases BTB permeability	miR-181d-5p	NEAT1	SOX5	tight junction proteins	[31]

**Table 2 cells-11-01670-t002:** The pathophysiological role of miR-181 in EC angiogenesis.

Function/Effects	MiR-181	Upstream	Target	Downstream Events/Signaling	Ref
suppress HMEC-1 angiogenic function	miR-181a	H19	MMP-2, MMP-9,	VRGF↓, eNOS↓, H19-activated JNK, and AMPK signaling↓	[46]
inhibits ocular neovascularization	miR-181a-5p		Bcl2, MAPK1	VEGF signaling↓	[56]
inhibits retinal neovascularization	miR-181a-5p		endocan	ERK signaling↓	[60]
promotes HUVEC angiogenesis	miR-181a		CaM	CaM-CaMKII pathway↓	[65]
suppress HUVEC angiogenesis	miR-181a/b-5p		PDGFRA		[48]
suppresses HUVEC angiogenesis	miR-181b		CCN1	AMPK signaling↓	[49]
suppresses arsenic-induced angiogenesis	miR-181b		NRP1		[50]
promotes rat BMEC angiogenesis	miR-181b-5p		TRPM7, TIMP3	HIF-1α↑, VEGF↑,	[68]
promotes BM-EPC angiogenesis	miR-181c-5p	SNHG1	SFRP1	Wnt3a/β-catenin signaling↑	[71]
promotes retinal pathological angiogenesis	miR-181c	CircPDE4B	VHL	HIF-1α, VEGF signaling↑	[74]
promotes angiogenesis in the endometrium	miR-181c	GAS5	TIMP3	VEGF signaling↑	[77]
suppresses HUVEC angiogenesis	miR-181d-5p	NEAT1	CDKN3	Akt signaling ↓	[80]

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
