# Peer review of "The Role of MiR-181 Family Members in Endothelial Cell Dysfunction and Tumor Angiogenesis"

_cells, 2022, doi:10.3390/cells11101670_

Round 1

Reviewer 1 Report

Although the Tables are indeed informative, I would like to encourage the authors to include 1-2 figures, describing the key functions and role of the miR-181 family members in aberrant angiogenesis.

Author Response

Review#1

Although the Tables are indeed informative, I would like to encourage the authors to include 1-2 figures, describing the key functions and role of the miR-181 family members in aberrant angiogenesis.

Response: As suggested by the reviewer, Figure 1 and Figure 2 were added in our manuscript which describing the key roles of the miR-181 family members in endothelial cell angiogenesis and in angiogenesis of different types of cancer respectively (Figure 1 on page 6 and Figure 2 on page 8).

Reviewer 2 Report

The review manuscript by Yang et al. summarized the literature reports on the roles of miR-181 family members in endothelial cell (dys)function and angiogenesis. The authors performed extensive literature search, and presented detailed information on miR-181 in EC function, angiogenesis and tumor in the 3 tables. In the last part of the manuscript, the authors discussed the potential use of lipid nanoparticles as a platform to deliver angiomiR for cancer therapy. Overall, the manuscript is well written and the literature coverage is comprehensive. There is, however, a major concern on the function of miR-181 in the referred experimental settings. As the authors noted, the results from prior research on miR-181 are often inconsistent or contradictory. MiR-181 can be pro- or anti-angiogenic; can either break down or be required for the maintenance of the blood-tissue barrier. Because this is a review manuscript, the authors will need to discuss the inconsistency and provide some of their own explanations. One option to clarify the controversies will be to separate the in vitro cell culture studies, which tend to be more variable and prone to artifacts, from findings from animal models, which are more relevant to the disease state. The conclusion of the current draft, that miR-181 family members may be master regulators of tumor angiogenesis, invasion, and metastasis (line 339), is not well supported. Before the targeting cells or mRNAs can be further defined, using lipid nanoparticles to deliver miR-181 mimics is unlikely to be a logical therapeutic approach.

Other specific suggestions:

1)      The review manuscript can be improved by adding illustrative figure(s). For example, the authors may consider a figure demonstrating the sequence homology and difference among miR0181 a-d, and their gene loci on different chromosomes, or some of the functions/targeting mRNAs mentioned elsewhere in the manuscript

2)      Line 145, SOX5 is not expressed in non-neoplastic tissues. Line 161, downregulation of SOX5 by miR-181c contributed to the dysregulation of tight junction in blood-spinal cord barrier. The statements are not consistent

3)      As listed in table 2, miR-181 can be either pro- or anti-angiogenic. Line 108, overexpression of miR-181a/b improved hESC-EC-induced neovascularization angiogenesis in a mouse model of limb ischemia. Line 179, overexpression of miR-181b suppressed angiogenesis in mouse hindlimb ischemia model. The statements are contradictory and will need further discussion.

4)      Similar to the controversies in angiogenesis, the involvement of miR-181 in cancer and tumor-related angiogenesis was not clear. Is miR-181 downregulated or upregulated in cancer? Is miR-181-containing exosome pro- or anti-angiogenesis?

Author Response

Review #2

Overall, the manuscript is well written and the literature coverage is comprehensive. There is, however, a major concern on the function of miR-181 in the referred experimental settings. As the authors noted, the results from prior research on miR-181 are often inconsistent or contradictory. MiR-181 can be pro- or anti-angiogenic; can either break down or be required for the maintenance of the blood-tissue barrier. Because this is a review manuscript, the authors will need to discuss the inconsistency and provide some of their own explanations. One option to clarify the controversies will be to separate the in vitro cell culture studies, which tend to be more variable and prone to artifacts, from findings from animal models, which are more relevant to the disease state. The conclusion of the current draft, that miR-181 family members may be master regulators of tumor angiogenesis, invasion, and metastasis (line 339), is not well supported. Before the targeting cells or mRNAs can be further defined, using lipid nanoparticles to deliver miR-181 mimics is unlikely to be a logical therapeutic approach.

Response: We highly appreciate the reviewer’s insightful comments and helpful suggestion on our manuscript. We have made corrections according to reviewer’s suggestions (seeing in the 3rd paragraph on page 6).

1)      The review manuscript can be improved by adding illustrative figure(s). For example, the authors may consider a figure demonstrating the sequence homology and difference among miR0181 a-d, and their gene loci on different chromosomes, or some of the functions/targeting mRNAs mentioned elsewhere in the manuscript

Response: An illustrative figure demonstrating the sequence homology and difference among miR-181a-d, and their gene loci on different chromosomes has been published and well presented by Indrieri  et al. in a review article [1].

As suggested by the reviewer, we provided Figure 2 (roles of the miR-181 family members in endothelial cell angiogenesis) and Figure 3 (key roles of the miR-181 family members in angiogenesis of different types of cancer) in this manuscript (Figure 1 on page 6 and Figure 2 on page 8).

2)      Line 145, SOX5 is not expressed in non-neoplastic tissues. Line 161, downregulation of SOX5 by miR-181c contributed to the dysregulation of tight junction in blood-spinal cord barrier. The statements are not consistent.

Response: We thank the reviewer for pointing out this error in line 145. We made correction in the revised manuscript.  In fact, SOX5 is found highly expressed in spermatids, neurons, oligodendrocytes, and chondrocytes.

3)      As listed in table 2, miR-181 can be either pro- or anti-angiogenic. Line 108, overexpression of miR-181a/b improved hESC-EC-induced neovascularization angiogenesis in a mouse model of limb ischemia. Line 179, overexpression of miR-181b suppressed angiogenesis in mouse hindlimb ischemia model. The statements are contradictory and will need further discussion.

Response: Regarding the contradictory statements of the role of miR-181a, we have provided more information for the two experiments mentioned in line 108 and line 179, respectively (seeing line line 109-113, and line184-186).

4)      Similar to the controversies in angiogenesis, the involvement of miR-181 in cancer and tumor-related angiogenesis was not clear. Is miR-181 downregulated or upregulated in cancer? Is miR-181-containing exosome pro- or anti-angiogenesis?

 Response: According to the studies of the involvement of miR-181 members in tumor angiogenesis, the role of each individual miR-181 family member in EC angiogenesis and expression level depends on the type of tumor or cell type. The functions of extracellular vesicles (EVs)-encapsulated miR-181 also depend on the type of parental cells.

Reviewer 3 Report

This review summarizes the current state of knowledge of the role of miR-181 family members in endothelial cell dysfunction, emphasizing on their pathophysiological roles in angiogenesis. The actions of miR-181 members are summarized concerning their targets and associated major angiogenic signaling pathways in a cancer-specific context. Elucidating the underlying functional mechanisms of miR-181 family members that are dysregulated in endothelial cells or cancer cells is important for developing miRNA-based therapies for angiogenesis-related diseases such as retinopathies, and cancer. Finally, potential clinical applications of miR-181 family members in anti-angiogenic tumor therapy are discussed.

Elucidating the underlying mechanisms of miR-181 family members that are dysregulated in endothelial cells or cancer cells is important for developing miRNA-based therapies for angiogenic diseases. This manuscript is a timely review on the important topic, which could expedite the appreciation of the therapeutic potential of miR-181s for patients with angiogenesis-related diseases.

Page 9, Lines 368-370: It is verbose to distinguish between virus-like nanoparticles and non-viral nanoparticles. They both can be referred to as nanoparticles.

Author Response

Review#3

Comments and Suggestions for Authors

Elucidating the underlying mechanisms of miR-181 family members that are dysregulated in endothelial cells or cancer cells is important for developing miRNA-based therapies for angiogenic diseases. This manuscript is a timely review on the important topic, which could expedite the appreciation of the therapeutic potential of miR-181s for patients with angiogenesis-related diseases.

Page 9, Lines 368-370: It is verbose to distinguish between virus-like nanoparticles and non-viral nanoparticles. They both can be referred to as nanoparticles.

Response: We appreciate the positive feedback from the reviewer. As suggested by the reviewer, we shorten this sentence.

Round 2

Reviewer 2 Report

The revised manuscript is much improved; the authors are highly responsive to the suggestions.